# Functional Polymer Materials for Advanced Lithium Metal Batteries: A Review and Perspective

**DOI:** 10.3390/polym14173452

**Published:** 2022-08-24

**Authors:** Ting Ma, Xiuyun Ren, Liang Hu, Wanming Teng, Xiaohu Wang, Guanglei Wu, Jun Liu, Ding Nan, Xiaoliang Yu

**Affiliations:** 1College of Chemistry and Chemical Engineering, Inner Mongolia University, Hohhot 010021, China; 2Inner Mongolia Key Laboratory of Graphite and Graphene for Energy Storage and Coating, School of Materials Science and Engineering, Inner Mongolia University of Technology, Hohhot 010051, China; 3College of Materials Science and Engineering, Qingdao University, Qingdao 266071, China; 4Department of Mechanical Engineering, Research Institute for Smart Energy, The Hong Kong Polytechnic University, Hong Kong 999077, China; 5Rising Graphite Applied Technology Research Institute, Chinese Graphite Industrial Park-Xinghe, Ulanqab 013650, China

**Keywords:** polymer, battery, review

## Abstract

Lithium metal batteries (LMBs) are promising next-generation battery technologies with high energy densities. However, lithium dendrite growth during charge/discharge results in severe safety issues and poor cycling performance, which hinders their wide applications. The rational design and application of functional polymer materials in LMBs are of crucial importance to boost their electrochemical performances, especially the cycling stability. In this review, recent advances of advanced polymer materials are examined for boosting the stability and cycle life of LMBs as different components including artificial solid electrolyte interface (SEI) and functional interlayers between the separator and lithium metal anode. Thereafter, the research progress in the design of advanced polymer electrolytes will be analyzed for LMBs. At last, the major challenges and key perspectives will be discussed for the future development of functional polymers in LMBs.

## 1. Introduction

The accelerating developments in power vehicles and portable electronic devices put forward higher requirements for the energy density of traditional lithium-ion batteries [1,2]. Recently, commercial lithium-ion batteries with graphite anode materials are facing theoretically extremely high values of energy density [3,4]. Therefore, electrode materials with high capacity need to be developed urgently. Among them, lithium metal is attracting considerable attention as a next-generation anode material owing to its ideal theoretical specific capacity (3860 mAh g^−1^) and ultra-low redox potential (−3.04 V versus standard hydrogen electrode) [5].

Despite the above advantages, lithium metal batteries face many challenges in the practical process. First, the high chemical activity of metallic lithium makes it easy to react irreversibly with most electrolytes, thence forming a solid electrolyte layer (SEI) on the anode surface [6]. The formation of SEI film consumes active lithium and reduces Coulombic efficiency. The formation of non-uniform SEI leads to an uneven diffusion of lithium ions, which affects the initial lithium nucleation and aggravates the formation of lithium dendrites. Second, due to the poor conductivity of the SEI film, the transport of electrons between Li dendrites will be blocked or even completely isolated; these blocked and deactivated lithium metals cannot be reused in the subsequent cycle of the battery, so they are called “dead lithium”. The formation of dead lithium increases internal resistance and polarization, and it further reduces the Coulombic efficiency [7]. Third, the irregular growth of lithium dendrites can pierce the separator easily, even leading to a series of safety problems such as battery short circuit. Finally, the continuous plating/stripping during the cycle leads to intense changes in the volume of the lithium metal, which may cause repeated rupture and formation of the SEI film, resulting in poor cycling stability. Therefore, it is highly essential for advancing the development of LMBs to prepare a safe and stable lithium metal anode [8,9,10,11].

Obviously, the interfacial problem of lithium metal anodes is the critical factor hindering their practical application. In this case, various efforts have been devised to optimize the interface stability of LMBs, which include: (1) electrolyte modification to adjust SEI composition [12,13,14]; (2) artificial SEI to enhance certain properties of SEI [15,16]; (3) separator modification [17,18,19]; (4) solid-state electrolyte [20,21]; and (5) 3D collector to accommodate volume changes during cycling [22,23]. For the construction of a stable interface, polymers play a significant role in LMBs because of the high flexibility as well as the low interface resistance with electrodes. For example, the mechanical properties of the polymer guarantee it to be a protective layer to prevent the uncontrolled growth of lithium dendrites; some polymers react with lithium metal to modulate the SEI composition and enhance certain properties of the SEI, etc. Because of the significant increase in research activity involving functional polymer materials design for LMBs, a timely and comprehensive review is urgent to examine this attractive and continuously evolving field of study. Novel functional polymer synthesis/design strategies proposed recently as well as a systematic classification are summarized to better demonstrate the progress of this research field.

In this review, we will discuss the most important roles of functional polymer materials in LMBs: polymeric artificial SEI, polymeric functional interlayer, and polymer electrolyte (Figure 1). Recent advances in the development of advanced polymeric materials are examined for boosting the stability and cycle life of LMBs as different components including artificial solid electrolyte interface (SEI) and functional interlayers between separator and lithium metal anode. Artificial SEI is a very thin artificially designed film with a good ionic conductivity as well as moderate mechanical strength and flexibility. A polymeric interlayer is a functional film placed between separators and lithium metal anodes. The thickness is generally higher than that of artificial SEI, normally in the range of tens of nanometers to several microns. Thereafter, we will review the research progress in the design of advanced polymer electrolytes for LMBs, which as alternatives of liquid electrolytes, have attracted extensive research attention because of their low flammability, flexible processability, and high resistance to mechanical deformation. The relationship between functionalized polymer materials and improved electrochemical performance will be elucidated. Finally, the development direction of polymer materials in LMBs in the future will also be prospected.

## 2. Polymeric Artificial SEI

Generally, in liquid lithium-ion batteries, a passivation layer can be deposited on the surface of electrode materials ascribing to the reaction with electrolyte during the first charge–discharge process. This passivation layer is an electronic insulator but an excellent conductor of Li^+^. Li^+^ can be embedded and extracted freely through the passivation layer, which is called “solid electrolyte interface film” (SEI) [24]. Compared with lithium-ion batteries, LMBs place higher requirements on SEI films. In addition to having the most basic ion conduction and electron-blocking capabilities, SEI films in Li metal batteries require more uniform composition, structure and ionic conductivity. At the same time, the volume of lithium metal surface changes significantly due to repeated deposition/stripping during the cycle, so the SEI film needs to have certain strength or good flexibility. Unfortunately, the poor interfacial stability of the SEI film formed spontaneously between the electrolyte and lithium metal affects the cycling performance of the battery [25,26].

To optimize the performance of lithium metal anodes, a lot of research studies have been devoted to constructing artificial SEIs. The physicochemical properties of artificial SEI films can be carefully designed and tuned to compensate for the shortcomings of naturally formed SEI films. Polymers are widely used in artificial SEI due to their good flexibility, certain ionic conductivity, and low interfacial resistance [27,28]. This section is divided into two subsections: polymer artificial SEI and inorganic/polymer composite artificial SEI.

### 2.1. Polymers as Artificial SEI

Polymer artificial SEIs are functional polymer or polymer composite coatings rationally designed to provide excellent interfacial properties. Polymer artificial SEI prepared by an ex situ method is easy to produce and directly coated on the electrode surface. Since the polymerization process does not chemically bond with the electrode, there are many options for materials and preparation methods for the coating. 

Polyethylene oxide (PEO) films are widely used in LMBs because of their non-reactivity with lithium, strong ability to modulate Li^+^ diffusion, and the electrical insulation properties. Addisu et al. applied polyethylene oxide (PEO) as an artificial SEI on the copper current collector for uniform lithium metal deposition, as shown in Figure 2a [29]. Li-ions interact electrostatically with the polar oxygen atoms in PEO to adjust the Li^+^ flux and guide uniform lithium deposition. The PEO film encapsulates the deposited lithium to shed lithium from electrolyte contact and promotes the production of a strong but thin SEI layer. Moreover, the PEO film interacts with the lithium salt (such as lithium bis(trifluoromethanesulfonyl) imide (LiTFSI)) in the electrolyte, so it can reduce the consumption of the electrolyte on the electrode interface and enhance the Li^+^ transfer. A uniform PEO thin film coating enables highly reversible lithium plating/stripping cycles. The assembled anode-free Cu@PEO||LiFePO_4_ full cell was determined to cycled stably, of which the average Coulombic efficiency and capacity retention attained 98.6% and 30% over 200 cycles at 0.2 °C. Similarly, Dong et al. applied 18-crown-6 ether and polyvinylidene fluoride PVDF coated onto the lithium anode surface as artificial SEI to suppress lithium dendrites [30]. PVDF is a flexible host, possessing high electrochemical stability and good dielectric properties. 18-crown-6 and PVDF composite coatings provide flexibility and high lithium-ion conductivity, which inhibits dendrite growth and improves cycling stability. Biopolymers are natural polymers produced from living organisms, which are resource-rich and environmentally friendly. Zhang et al. employed natural agarose (AG) film, which is ionically conductive, highly elastic and stiff, and chemically stable, to coat the copper current collector [31]. Thanks to the tough and elastic features of agarose polymer film, the excessive volume change can be relieved during the Li plating/stripping process, thereby impeding the Li dendrites from its overgrowth. The high ionic conductivity enables fast Li-ion transportation. Similarly, alginate biopolymer [32] and chitosan [33,34] have also been reported as functional modification layers on current collectors.

Fan et al. coated a layer of lithophilic polythiophene derivative (PTs) matrix on copper foil to realize a uniform deposition of lithium metal [35]. According to the first-principles simulation, polythiophene derivatives with shorter side chains are more profitable due to their higher adsorption energy and diffusion rate compared with lithium ions (Figure 2c). There are numerous dispersed lithophilic sites that support lithium-ion diffusion channels on the backbone of the polythiophene derivative substrate, contributing to the uniform nucleation and growth of lithium ions. The authors concluded that polythiophene derivatives that have diverse side chains enable lithium ion electrodeposition into different morphologies, as shown in Figure 2b. 

The application of a highly viscoelastic polymer on the lithium metal electrode may decrease the local current density and make a more uniform morphology of deposited lithium. Cui et al. coated the electrode with a soft and highly viscoelastic polymer to achieve uniform lithium deposition [36]. This polymer with mobility flows over the electrode surface to repair SEI fractures and pinholes generated during lithium plating/stripping. When repairing SEI defects, the ionic conductivity of the polymer itself is lower than that of the liquid electrolyte, which can reduce the local current density and prevents the growth of lithium dendrites. At a current density of 1 mA cm^−2^, the modified electrode exhibited a stable Coulombic efficiency of about 97% over 180 cycles. Likewise, in Liu’s previous work, a dynamic cross-linked polymer silly putty (SP) was employed as a lithium metal anode artificial SEI layer (Figure 2d) [37]. The “solid–liquid” feature of SP is reflected in Figure 2e. SP not only adapts to electrode volume change in the process of lithium metal deposition and stripping but also reduces the side reaction between electrolyte and lithium by restricting electrolyte from lithium anode. More importantly, the rigidity and fluidity characteristic of this kind of dynamic polymer can be reversibly switched according to the growth rate of lithium to uniformly cover and prevent the dendrites growth. 

Polymers with microporous structures can increase the number of Li^+^ transport to reduce the polarization in LMBs. Ma et al. used a polymer with intrinsic micropores (PIM) to coat lithium foil [38]. The liquid electrolyte is confined in a porous polymer host, which increases its cation transport number because the length of its pores is similar to that of the Debye shield in the electrolyte, creating a diffusing electrolyte bilayer at the polymer–electrolyte interface retaining counteracting ions and repelling co-ions from the electrolyte due to its large size. Since Li-ion is the smallest ion in the system and has almost no interaction with this microporous polymer host, the diffusion of Li-ion in the pore network is higher than that of anions. As a result, the cation transport number increased significantly. This is followed by a reduction in battery polarization during lithium metal plating. In the assembled Li-Cu half-cell, the Coulombic efficiency of the modified electrode was >98% over 80 cycles at 0.4 mA cm^−2^.
Figure 2(**a**) Schematic diagrams of Li plating and stripping. Lithium deposited on (top) bare copper electrodes (middle) micro-PEO thin film coating (bottom) with optimized nanoscale PEO thin film coating [29]. Reproduced with permission. Copyright 2018, Royal Chemical Society. (**b**) Schematic diagram of plating uniform lithophilic PTs substrates on Cu foils. (**c**) (Left) Chemical formula of PTsO1, (Right) chemical formula of PTsO4 [35]. Reproduced with permission. Copyright 2021, Elsevier (**d**) Molecular structure of SP. (**e**) Digital pictures of SP mounted over a hole punched on a Petri dish. (**f**) Two front views at different time points [37]. Reproduced with permission. Copyright 2017, American Chemical Society.
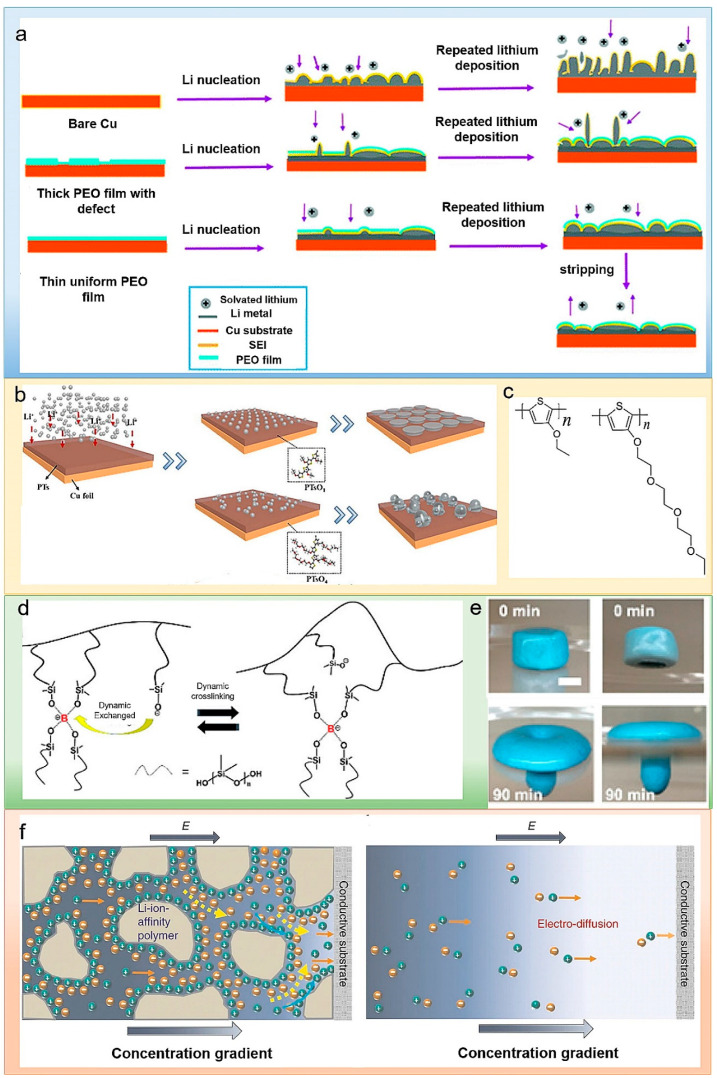



Compared with artificial SEI constructed by ex situ methods, the in situ counterparts are chemically bonded to the electrode, and the in situ SEI layer is tightly connected to the lithium metal anode, which is helpful for the better mechanical performance and interfacial compatibility of the electrode. 

Gao et al. reported a novel skin-grafting method to regulate lithium deposition by coating the surface of lithium metal with poly((N-2,2-dimethyl-1,3-dioxolan-4-methyl) base)-5-norbornane-exo-2,3-dicarboximide)), a chemically and electrochemically activated polymer layer, as shown in Figure 3a [39]. The layer consists of a ring ether group with a rigid polycyclic skeleton as a grafted polymer surface layer on metallic lithium anodes. It not only enables the incorporation of ether-based polymer components into SEI but also regulates Li plating/stripping performance in a dendrite-free manner. As a result, the Li/Cu half-cell delivers an average CE of 98.3% over 200 cycles at the current density of 0.5 mA cm^−2^ (Figure 3b). 

The abundant polar functional groups in the polymers can manipulate the lithium ion flux and guide the uniform deposition of lithium. It could also facilitate the production of favorable components in SEI. As shown Figure 3c, an ultrathin new polyurea film protective layer was fabricated by Sun et al. [40]. The polyurea film can restrict Li deposition under the film and inhibit dendritic growth as a protective layer. There is a strong interaction between the nitrogen-containing polar groups in polyurea with lithium, thence modulating the lithium ions flux, reducing polarization and leading to uniform lithium deposition effectively (Figure 3d). Similarly, Luo et al. used a highly polar *β*-phase PVDF coating as an artificial SEI layer for dendrite-free lithium metal anode [41]. 

Some polymers react with lithium to produce functional artificial SEI-protected lithium anodes. Li et al. modified lithium foils by immersing them in polydimethylsiloxane (PDMS)-containing DOL/DME solutions [42]. Pure PDMS is not a lithium ions conductor, but the new protective layer (PDMS-LI) facilitates the conduction of lithium ions. An et al. polymerized tetramethylenevinylene (TME) in situ on the lithium metal surface for uniform lithium deposition [43]. Lithium ions on the lithium metal surface can form a complex with the C=C double bond of TME, producing a strong artificial SEI layer. Young’s modulus of the TME–lithium anode surface is increased to 16–18 GPa after TME modification. The high modulus inhibits the growth of dendrites and thus improves the cycling stability. Polydopamine (PDA) is known for its strong adhesion to substrates of any type and shape; it can functionalize the surfaces of any material. He et al. reported a functional PDA layer coated on Cu foil [44]. As shown in Figure 3e, lithium-coordinated carbonyl groups formed by lithium ions and hydroxyl groups on PDA serve as nucleation sites. These nucleation sites are conducive to the uniform nucleation of lithium metal and impede the formation and growth of dendrites. The Li-Cu half-cell delivered an average Coulombic efficiency of 97% for 100 cycles, and the symmetrical cell displayed an ultra-long cycle life of more than 800 h (Figure 3f). Similarly, Li et al. adjusted the deposition of lithium through an artificial SEI flexible layer formed via the in situ reaction of metal lithium and polyacrylic acid (PAA) [45]. The artificial SEI possesses the excellent binding ability and high stability LiPAA polymer, which can reduce the side reactions between lithium metal and electrolyte, thereby enhancing the cycling stability. 

Wang et al. introduced 3-methacryloxypropyltrimethoxysilane (MPS) to enhance the interfacial adhesion between Li metal and SEI films through chemical bonding and physical entanglement effects (Figure 3h) [46]. The MPS layer is tightly adhesive to the surface of the lithium anode through Li-O-Si interaction, effectively establishing a dense barrier to prevent the corrosive effect of air (Figure 3g). A lithium metal anode modified with MPS displayed outstanding electrochemical performance. The cycle stability of symmetric cells MPS-Li||MPS-Li reached 1400 h (current density, 1 mA cm^−2^). The specific capacity of MPS-Li||S cells exhibited still 652 mAh g^−1^ after 300 cycles at 0.2 °C.
Figure 3(**a**) (Left) Illustration of the different interfacial chemistries of Li metal with grafted skin in a carbonate electrolyte. (Right) Chemical structure of the polymer skin, composed of a cyclic ether group (pink) and a polycyclic main chain (blue). (**b**) CEs of Li deposition/dissolution on a Cu electrode. Reproduced with permission [39]. Copyright 2017, American Chemical Society (**c**) Schematic diagrams showing the effect of MLD coating on a Li anode when cycling. (**d**) Surface morphologies for bare Li and Li@P10 after 20 cycles of stripping/plating. The lengths of scale bars are 20 µm for images [40]. Reproduced with permission. Copyright 2019, John Wiley and Sons. (**e**) Diagram of the mechanism of PDA induced Li deposition. (**f**) Voltage–time profiles of the Li plating/stripping process with a cycling capacity of 0.2 mAh cm^−2^ at 0.1 mA cm^−2^ in symmetric Li@Cu/Li@Cu cells with pristine Cu and PDA-Cu current collectors (top). Long-term cycling performance of the LiFePO_4_/Li@Cu and the LiFePO_4_/Li@PDA-Cu cells at 0.5 C (bottom) [44]. Reproduced with permission. Copyright 2019, Elsevier. (**g**) The dense MPS layer isolates the lithium from the air, and the MPS layer protects the lithium metal (top). Schematic diagram of the exposure of raw Li to air (bottom). (**h**) Schematic illustration of the structure of MPS-Li [46]. Reproduced with permission. Copyright 2021, John Wiley and Sons.
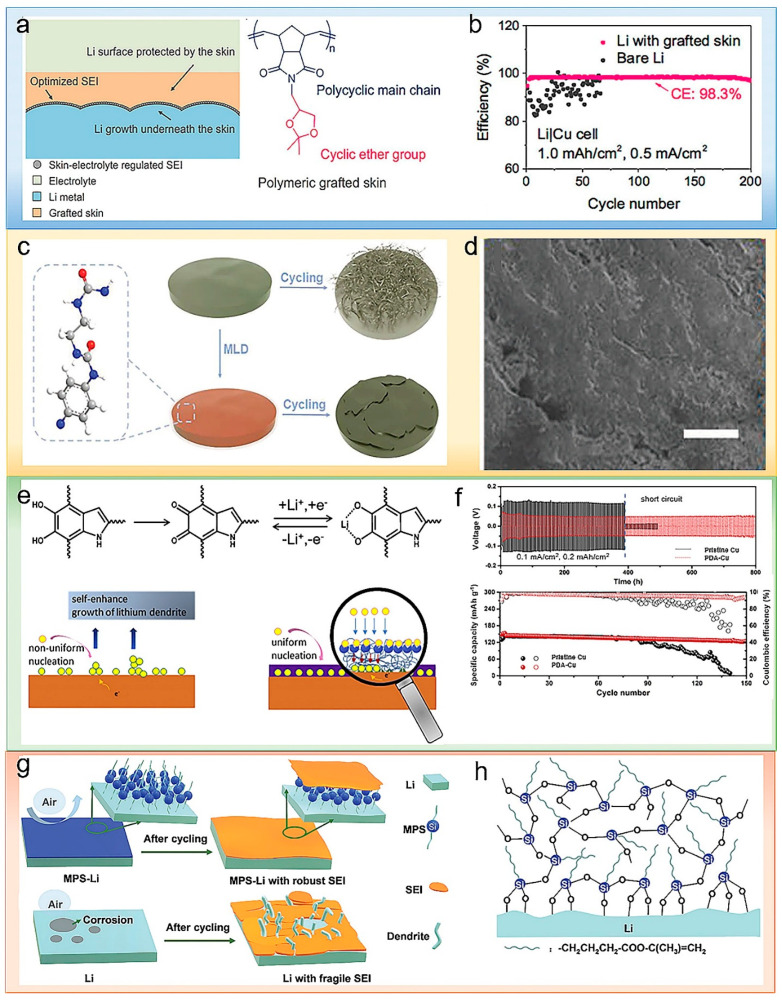



### 2.2. Inorganic/Polymer Composites as Artificial SEI

Inorganic additives themselves have high ionic conductivities, but usually, their flexibilities are poor. The SEI composed of inorganic filler and polymer components can integrate the advantages of both, showing high lithium ion conductivity, excellent mechanical strength and flexibility.

Inorganic materials with high lithium-ion conductivity are incorporated into polymers to improve overall ion conductivity. Yang et al. applied an artificial polymer/ceramic hybrid SEI film as a protective layer for lithium anode, which is composed of a garnet-type Li_7_La_3_Zr_1.75_Nb_0.25_O_12_ (LLZNO) ion conductor and a polyethylene oxide (PEO)-based polymer electrolyte [47]. The flexibility of PEO enables the hybrid film to buffer large lithium volume changes. PEO with ceramic LLZNO particles decoration remarkably improved their mechanical strength, which inhibits the growth of Li dendrites. As a solid ion conductor, LLZNO can dissociate additional Li-ions during interaction with anions and increase the flux of lithium ions, leading to Li deposition under the passivation layer. An assembled symmetrical cell with modified anode presented stable long cycling for 1000 h. The LiFePO_4_||PEO-garnets@SS full cell delivered excellent Coulombic efficiency of 99.5%. 

The surface diffusivity of Li ions affects the morphological structure of electrodeposited lithium. Halogenated compounds are generally accompanied with high surface diffusivity. A lithium metal anode protecting layer composed of LiF/PEO nanocomposite was proposed by Fu et al. [48]. The composite coating can block the contact between lithium anode and electrolyte to reduce electrolyte decomposition. LiF offers a low surface diffusion potential barrier and good electrochemical stability for lithium ion. PEO provides high donor numbers for lithium ion, which promotes the diffusion of lithium ions at the interface of LiF/PEO. When contacting with carbonate-based liquid electrolyte, the ionic conductivity and Li ion transfer number of the composite coating exhibited extremely high values of 0.97 mS cm^−1^ and 0.77, which facilitated uniform lithium deposition and suppressed the formation of lithium dendrites (Figure 4a,b). Therefore, the lithium metal anode with a protective layer coating can cycle above 1000 h at 1 mA cm^−2^ stably. Similarly, Sun et al. developed a thin artificial SEI layer with a LiF/defluorinated polymer composite [49]. The LiF/defluorinated polymer composite layer is constructed by a rolling process between polytetrafluoroethylene (PTFE) and lithium metal, as shown in Figure 4c. It can be observed in Figure 4d that the composite layer consists of the inner and outer layers with lithium (LiF-rich) and polyene/fluoropolymer (polymer-rich), respectively. The cavities among LiF particles can be counteracted by the polymer layer to improve the uniform coverage of the protective layer on the lithium metal surface. Similarly, Kim et al. prepared a protective layer composed of poly(3,4-ethylene dioxythiophene)-co-poly(ethylene glycol) (PEDOT-co-PEG) and AlF_3_ particles as an artificial SEI on lithium metal to stabilize the lithium electrode [50]. 

The high Li^+^ ion mobility number ensures the selective transport of cations for the uniform deposition of Li metal. A multifunctional artificial SEI was reported by Yu and his coworkers by a dynamic single-ion conductive network (DSN) [51]. The structure of DSN is shown in Figure 4e. The as-prepared DSN was assembled by a Al(OR)^4−^ (R = soft fluorinated linker) tetrahedral center as a dynamically bounding structure and counter anion, giving it mobility and Li^+^ single-ion conductivity, as shown in Figure 4f. At the same time, the fluorinated linkers endow DSN with chain fluidity and electrolyte-blocking ability, which inhibits the parasitic reactions between electrolyte and lithium metal. Benefiting from above advantages, the Li-Cu half-cell using commercial carbonate-based electrolytes exhibited stable cycling capability for more than 400 cycles.

High ionic conductivity is one of the key parameters for desirable SEI films. Adding certain metal compounds into polymer matrices can improve their ionic conductivities. Cui et al. reported an artificial SEI layer that includes Cu_3_N nanoparticles tightly bounded with styrene–butadiene rubber (SBR) (Figure 5a) [52]. The reaction between Cu_3_N and lithium metal induces the formation of a Li_3_N passivation layer with high Li^+^ conductivity. It can promote the Li-ions transportation across the surface of electrode effectively, resulting in uniform Li-ion flux. SBR provides good flexibility and ensures the integrity of the structure during lithium plating/stripping. Because of the excellent Li-ion conductivity and mechanical properties of the artificial SEI, the stability of the lithium metal anode is significantly improved. The SEM of Cu_3_N + SBR protecting Li metal is shown in Figure 5b. Hu et al. further reported the N-containing inorganic interface layer as an artificial SEI, which was fabricated via the in situ polymerization of *α*-ethyl cyanoacrylate with lithium nitrate as an additive [53]. It can promote ion conduction and block further adverse interface reactions. Tu et al. reported a perfluorinated ionomer and nanoporous Al_2_O_3_ composite as an artificial SEI to protect lithium metal anodes [54]. The ionomer owns a high ionic conductivity, which is further enhanced by the addition of nanoporous Al_2_O_3_. Adding Al_2_O_3_ in the artificial SEI can improve the mechanical strength, inhibiting the overgrowth of dendrite and improving cycling stability. Pang et al. used *β*-Li_3_PS_4_ (LPS) and polydimethylsiloxane (PDMS) composite hybrid films as an artificial SEI to protect the lithium metal anode, as shown in Figure 5c [55]. The *β*-Li_3_PS_4_ phase provides the Li^+^ transport channels to improve lithium-ion conductivity. Elastomeric polymers (PDMS) filled into the particle interstices increase the flexibility of the film to accommodate the deformation phenomenon during lithium metal plating/stripping. The Coulombic efficiency of LPS–PDMS–coated Cu is relatively stable, as shown in Figure 5d.

Lithium affinity is attributed to the wettability of the surface of lithium, and the addition of some metal ions in the polymer can increase the affinity. Meng et al. developed a polydopamine–copper ion (PDA-Cu^2+^) coating formed by particles aggregated with copper ions and nanoparticles [56]. The PDA coating presents a unique porous structure composed of sticky nanoparticles that can effectively regulate the deposition of lithium metal and increase the wettability of the electrolyte. Li ions can rapidly migrate through the gaps between the PDA nanoparticles due to the repulsive effect between similar charges. In this case, the modified lithium metal anode achieved great Coulombic efficiency and long-term stability. 

The alloy phase is not only rigid enough to prevent dendrite growth but also has a strong affinity for lithium. Jiang et al. fabricated a poly(tetramethylene ether glycol) (PTMEG)-Li/Sn alloy hybrid layer on the lithium metal anode surface [57]. The synthesis process involves simply dipping lithium metal into a 0.1 M solution of tin tetrachloride in tetrahydrofuran (THF). SnCl_4_ was directly reduced by Li metal upon contact, forming an alloy layer. At the same time, LiCl as a nucleation site and SnCl_4_ as a strong Lewis acid induce the ring-opening polymerization reaction of THF. The PTMEG-Li/Sn alloy hybrid layer provided a rapid lithium-ion transport channel as well as a strong affinity for lithium. As a consequence, the symmetrical cell is stably cycled for 1000 h at a current density of 1 mA cm^−2^ without Li dendrite formation. Similarly, Ma et al. created an artificial SEI through an in situ reaction of AlI_3_, lithium metal and dioxolane (DOL), which induces the formation of a surface coating including Li-Al, LiI and a thin polymer film [58]. Among them, LiI enabled the diffusion energy barrier reduction in lithium ions, Li-Al alloys provide an effective barrier for lithium dendrite formation, and thin polymer films can inhibit the side reaction between electrolyte and lithium metal. Similarly, Zhang et al. designed a functional hybrid coating layer containing LiF, lithophilic Li-Sb alloy, and flexible polymer styrene–butadiene rubber (SBR) [59]. Lithophilic Li_3_Sb alloys can provide fast transport channels for lithium ions. 

Artificial SEI reacts in situ to produce a variety of favorable decomposition products at the interface to enhance SEI. Gao et al. proposed to use reactive polymer composites poly(vinylsulfonyl fluoride-ran-2-vinyl-1,3-dioxolane) (P(SF-DOL))-graphene oxide nanosheets as an artificial SEI layer [60]. The presence of LiF can be seen in the TEM of Figure 5f. This multi-component artificial SEI is dense enough to prevent electrolytes from contacting with the Li anode and relieve side reactions between electrolyte and electrode, as shown in Figure 5e. Graphene nanosheets provide mechanical strength to hinder dendrite growth and increase the cycle stability of LMBs. The interfacial fluctuations could be reduced during Li deposition by 3D hosts. As a result, the average efficiency was as high as 99.1% at a current density of 2.0 mA cm^−2^ over 300 cycles. Similarly, Zhao et al. reported a multifunctional cross-linked membrane of sulfur-containing hybrid lithium polyphosphate and poly(2-chloroethyl acrylate) (PECA) to protect lithium metal anodes [61]. The film can promote the formation of a stable organic/inorganic hybrid SEI layer including lithium salt, inorganic Li_3_PS_x_, lithium sulfide, and polymer-based organic polysulfide on the lithium metal anode. Benefiting from this hybrid robust SEI layer, Li metal anodes achieved stable cycling without Li deposition dendrite.
Figure 5(**a**) Schematic illustration of the fabrication of the Cu_3_N + SBR composite artificial SEI. (**b**) SEM image and digital photography of the Cu_3_N + SBR protected Li metal foil [52]. Reproduced with permission. Copyright 2020, John Wiley and Sons. (**c**) Schematic illustration of the designed structure of the hybrid films coated on Li or Cu foils and optical photographs of the films. (**d**) The CE evolution of Li plating/stripping on the bare Cu and LPS-PDMS–coated Cu (Cu–LPS-PDMS) over cycling (1 mA cm^−2^,1 mAh cm^−2^) [55]. Reproduced with permission. Copyright 2018, National Academy of Sciences of the United States of America. (**e**) The RPC layer first forms a dense layer (red) by chemically passivating Li surface products. The attached RPC subsequently forms the SEI layer (green). Unreacted RPC (yellow). (**f**) TEM images of the interface of RPC-stabilized Li [60]. Reproduced with permission. Copyright 2019, Springer Nature.
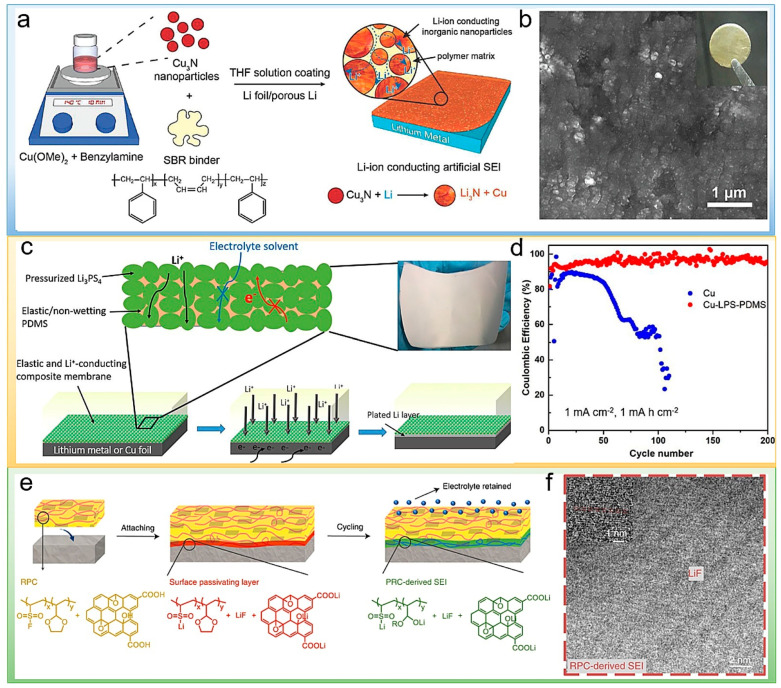



## 3. Polymeric Functional Interlayers in LMBs

Different from artificial SEI layers, which are normally attached or bonded on lithium metal anodes or current collectors, polymeric functional interlayers are placed between the separator and Li metal anode. They normally present larger thickness than artificial SEI. In this section, based on the composition, the functional interlayers are categorized into three types, i.e., single-polymer interlayer, multipolymer interlayer and inorganic/polymer composite interlayer.

### 3.1. Single-Polymer Interlayers

Electrospinning is a method developed in recent years to prepare one-dimensional nanofibers, which is widely used to prepare electrode materials or electrode material carriers for lithium batteries. Cui et al. designed a novel electrode by putting polyacrylonitrile (PAN) nanofiber with a three-dimensional (3D) network above the current collector [62]. The electrically insulating polymer may play the role as a 3D scaffold to lead the deposition of lithium on the surface of polymer fibers and in 3D polymer layers due to their surface polar functional groups with strong affinity for metallic lithium, as shown in Figure 6a. The Coulombic efficiency and areal capacity reaching 97.4% and 1 mAh cm^−2^ was achieved at a current density of 3 mA cm^−2^. as shown in Figure 6b. Similarly, a 3D nanofiber network composed by polyvinyl alcohol (PVA) was fabricated to modify current collector to modulate the uniform deposition of lithium by Wang et al. [63].

In previous work of Zhang et al., an electrospinning polymethyl methacrylate (PMMA) grafted with 4-amino-2,2,6,6-tetramethyl-1-piperidinyloxy (TEMPO) and 9-anthracene carboxylic acid (An) on a polypropylene (PP) separator was utilized as a radical functionalized bilayer porous composite membrane. The lithophilic functional groups of TEMPO contribute to the uniform nucleation and distribution of Li metal, and the cross-linked An groups help to enhance the mechanical behavior [64]. Similarly, Hwang et al. reported an electrospinning strategy to fabricate a three-dimensional network of ferroelectric *β*-phase polyvinylidene fluoride (*β*-PVDF) nanofibers supported on copper foil (PVDF#Cu) [65]. This structure not only facilitates the diffusion of Li ions through the electrolyte channels between *β*-PVDF fibers but also ensures transport along the ferroelectric surface by Li ion hopping on the fluoride side of PVDF. Furthermore, the 3D framework provides space to buffer volume changes during lithium plating/stripping.

Fan et al. plated lithium onto 3D porous poly-melamine-formalde-hyde (PMF) to optimize the cycling stability of LMBs [66]. Specifically, the 3D porous structure can accommodate volume changes during Li metal plating/stripping. PMF with plentiful N-containing polar groups has strong interactions with lithium ions, which result in the uniform distribution of lithium ions and regulation of lithium deposition.

Pan et al. covered both sides of polyethylene separators with porous cellulose nanolayers [67]. The mesoporous cellulose layer can achieve uniform lithium-ion flux and improve the cycling stability of LMBs. Meanwhile, the nanocellulose layer exhibits good electrolyte wettability and thermal stability. The modified separator does not deform at 200 °C, which can effectively prevent battery thermal runaway. As a result, a high capacity retention rate of 97.5% can be obtained after 65 cycles in LiFePO_4_||Li half-cell with the modified separator.

### 3.2. Multipolymer Interlayer

Bae et al. designed a polar polymer network (PPN) protective layer formed by the thermal cross-linking of polyacrylonitrile (PAN) and polyethylene glycol diacrylate (PEDGA) [68]. This PPN protective layer could inhibit highly corrosive cyclic carbonates by modulating the polymer–solvent interaction. In the PPN, the reactivity of C=O groups of the carbonate solvent can be effectively reduced by C≡N groups of the PAN polymer chains, resulting in the formation of an SEI layer with higher inorganic components. The PPN layer effectively inhibits the generation and growth of Li dendrites, thereby improving the Coulombic efficiency and prolonging the cycle life.

Ryu et al. fabricated a ferroelectric terpolymer–polydopamine interlayer (f-PTC-PD) [69]. f-PTC with high ferroelectricity is beneficial to enhance ionic conductivity and transference number, which achieves the uniform transport of Li-ion. The protective PD layer suppresses the dissolution of PTC and improves the overall affinity for liquid electrolytes. The schematic diagram of the f-PTC layer protecting Li metal is shown in Figure 7a. The f-PTC layer promotes the non-dendritic deposition of Li metal. The Li||LiFePO_4_ full cell constructed with an f-PTC-modified separator displayed a capacity retention rate of 84.4% after 100 cycles, and the SEM morphology was flat, as shown in Figure 7b. Similarly, Wang et al. fabricated a hybrid poly(dopamine)/octaammonium POSS (PDA/POSS) coating on the surface of PE separators [70]. This PDA/POSS coating imparted different surface characteristics to the PE separators while maintaining their microporous structure. These optimized surface features contributed not only to enhanced lithium-ion transfer number and ionic conductivity of the PE separator but also improved lithium/electrolyte interfacial stability, which effectively reduces electrode polarization and inhibits the formation of lithium dendrite, achieving great rate capability and cycling performance. Likewise, Zhang et al. uniformly deposited functionalized lithophilic polymers consisting of chitosan (CTS), polyethylene oxide (PEO), and polytriglycol dimethacrylate (PTEGDMA) onto a commercial Celgard separator through electrospray and polymer photopolymerization techniques [71]. The lithophilicity of the CTS-PEO-PTEGDMA layer ensured homogeneous lithium deposition and promotes a robust uniform SEI layer that suppressed the growth of lithium dendrites, generating an enhanced electrochemical performance of the lithium metal battery.

Yang et al. synthesized an adaptive buffer layer (ABL), which is composed of polypropylene carbonate, polyethylene oxide (PEO), and lithium salt, to regulate the interface between metallic lithium and solid electrolyte (Figure 7c) [72]. ABL owns good viscoelasticity (Figure 7f–i), so it can accommodate the shape change of lithium metal anode and ensure the intimate contact between lithium metal and solid electrolyte during cycling (Figure 7d,e). With ABL working in the lithium metal anode, the interface resistance of the Li/ABL/SPE/LiFePO_4_ battery enhanced by merely 20% over 150 charge/discharge cycles, while the counterpart without ABL increased by 117%. After that, Subramani et al. prepared a networked solid-state polymer (NSP) as an artificial interlayer to address lithium dendrite issues in the lithium metal battery [73]. The designed interlayer consists of poly(ethylene oxide-co-epoxypropane) (P(EO-co-PO)), poly(dimethylsiloxane) diglycidyl ether (PDMSDGE) chains and bis(trifluoromethanesulfonyl) lithium imide salts. The methyl groups on the P(EO-co-PO) and PDMSDGE chains reduce the surface energy of the interlayer, which can completely cover the high-energy lithium metal and modulate the transport of lithium ions. Lower surface energy is more favorable to the generation of Li-F bonds, which promotes Li uniform deposition. Furthermore, the elastic PDMS chains enable the NSP to buffer the lithium volume change during cycling. All the above advantages of NSP contributed to long-term cycling stability and improved Coulombic efficiency.

### 3.3. Inorganic/Polymer Composite Interlayer

Liu et al. developed a porous conductive interlayer (PCI) consisting of carbon nanofibers (CNF) and poly(m-phenylene dicarboxamide) (PMIA) to impede the growth of lithium dendrites in lithium metal anode [74]. PCI owns superb electrical conductivity provided by CNF and the excellent mechanical properties from PMIA. When lithium metal comes into contact with PCI during cycling, an equipotential surface is formed at their interface, which is conducive to eliminate the tip effect on lithium metal and realize uniform lithium deposition.

Wu et al. fabricated a functional porous double-layer composite separator to regulate uniform lithium deposition [75]. The modified separator was obtained via blade coating the polyacrylamide-grafted graphene oxide molecular brush on a commercial polypropylene separator (GO-g-PAM@PP). The schematic diagram of Li deposition on the GO-g-PAM@PP separator electrode is shown in Figure 8a. Graphene oxide in molecular brushes offers mechanical strength and fast electrolyte diffusion channels. The polyacrylamide chain contains a huge number of polar N-H and C=O bonds, providing functional sites for the effective adhesion and uniform distribution of lithium ions. As a consequence, dendrite-free lithium deposition with excellent Coulombic efficiency and cycling stability was achieved, as shown in Figure 8b. Xu et al. hybridized poly(vinylidene-co-hexafluoropropylene) (PVDF-HFP) and LiF into composite films as interlayers between Li metal anodes and electrolyte [76]. The good shape consistency obtained from the PVDF–HFP matrix and high Young’s modulus conferred by LiF bring various favorable characteristics, such as high mechanical strength, good ionic conductivity, excellent shape compliance, and good lithium metal anodes compatibility. All of the above merits lead to dendrite-free and dense Li deposition as well as reduced interfacial resistance.

Huo et al. coated a polypropylene (PP) separator with a hierarchical porous interlayer composed of Li_6.4_La_3_Zr_1.4_Ta_0.6_O_12_ (LLZTO) and polyvinylidene fluoride (PVDF) to regulate the uniform deposition of Li^+^ [77]. The Lewis acid–base interaction between LLZTO and PVDF led to a superb ionic conductivity and high Li^+^ transportation. The 3D fast Li^+^ transport pathways in the coating interlayer can efficiently redistribute the inhomogeneous Li^+^ flux from the insulating PP separator. Additionally, the interlayer could anchor the anions and guide a flexible Li^+^ transportation, achieving uniform deposition on the lithium anode (Figure 8c). All those advantages contribute to higher Coulombic efficiency and improved cycling stability. Similarly, Yan et al. designed a mixed ion/electronically conductive interlayer formed in situ from an intermediary Mg_3_N_2_ layer modified on polyethylene oxide (PEO) to tune Li deposition. PEO as a framework also provides lithium-ion transport [78]. The Mg_3_N_2_ layer is converted to Li_3_N and metallic magnesium through an in situ electrochemical reaction with the lithium anode. Li_3_N as a fast Li^+^ conductor can help to achieve fast transport of Li^+^ and alleviate the Li^+^ concentration gradient. Mg metal is beneficial to the uniformity of current density distribution during cycling. Therefore, this interlayer avoided the creation of Li dendrites and improves the stability and kinetics of the lithium anode interface.

In summary, functional polymer interlayers in LMBs play critical roles in regulating Li^+^ flux toward dendrite-free Li deposition. The thicknesses of interlayers are generally in the range of tens of nanometers to several microns, as summarized in the following Table 1. In contrast, polymer electrolytes possess much higher thickness, normally over tens of microns, which will be discussed in detail in the following section.

## 4. Polymer Electrolyte

Polymer electrolytes have attracted extensive research attention since their discovery. They demonstrate favorable flexibility, desirable Li^+^ ion conductivity, and good compatibility with both cathode and lithium metal anode [79]. In this section, the advancement of polymer electrolytes will be summarized by three categories, i.e., salt-in-polymer electrolyte, polymer-in-salt electrolyte, according to the various ion migration mechanism. The key difference between the former two is the lithium salt content, which are less than 50% and larger than 50%, respectively.

### 4.1. Salt in Polymer Electrolyte

Wu et al. fabricated an ultra-thin and flexible solid-state electrolyte (SSE), which was achieved by immersing a 5 µm thick separator into polyethylene oxide (PEO) and lithium bis(trifluoromethylsulfonyl) (LiTFSI) polymer electrolyte [80]. The strong and flexible polyethylene separator maintains the matrix of the polymer electrolyte, preventing internal short circuits and the penetration of dendrites. Similarly, Wang et al. synthesized a vertically aligned Li_1.5_Al_0.5_Ge_1.5_(PO_4_)_3_ (LAGP)/polyethylene oxide (PEO) composite solid-state electrolyte, with vertical LAGP walls providing continuous pathways for rapid ion diffusion [81]. Similarly, Pan et al. used (3-chloropropyl)trimethoxysilane (CTMS) as a bridging agent to chemically bond Li_10_GeP_2_S_12_ (LGPS) with polyethylene glycol (PEG) [82]. The polyethylene oxide (PEO) and bis(trifluoromethanesulfonyl)lithium imide (LiTFSI) were added to form uniform hybrid solid electrolytes (HSEs). The schematic structure of the HSEs is shown in Figure 9a. The large amount of -OH in PEG can promote the bonding with CTMS, leading to improved ionic conductivities and increased Li^+^ transference numbers. The strong chemical bonds tightly bonded ceramic and polymer, thereby solving the problem of interfacial compatibility. As a result, the LTO|PEO/PEG-3LGPS|Li battery delivered a high capacity retention rate of 94% after 120 cycles, as shown in Figure 9b. Furthermore, the cycling stability of all-solid-state Li|PEO/PEG-3LGPS|LiFePO_4_ (LFP) battery exhibited good performance at a high temperature of 60 °C (Figure 9c). Zhou et al. fabricated a new gel polymer electrolyte, and poly(ethylene glycol) methacrylate (PEGMA) contains glycol chains which can improve ionic conductivity [83]. The acrylate functional group of dipentaerythritol penta-hexa-acrylate (DPEPA) is capable of forming a highly cross-linked gel network. Ionic liquids with fluorinated side chains effectively immobilize the lithium salt anion by non-covalent interaction with the Lewis acid site, which reduces the affinity of lithium for the oxygen atom of the glycol chain and increases the lithium-ion transport number, finally improving the electrochemical stability of the polymer electrolyte.

Poly(ethylene oxide) (PEO) and poly(N-methyl-malonic amide) (PMA) were employed to make a double-layer composite polymer electrolyte (DLPSE) by Zhou et al. [84]. Figure 9d showed the PMA molecular structure and stacking model of DLPSE in an all-solid-state cell. PMA has a high dielectric constant but reacts easily with lithium metal, so the PMA side only came into contact with the cathode, which ensures extracting Li^+^ from the cathode without oxidizing the electrolyte at high voltage. Meanwhile, the PEO side is close to the lithium metal, providing a dendrite-free deposition of lithium. The polymer electrolyte layer performed good adhesion and high flexibility (Figure 9e,f), which improves the solid electrolyte/electrode interface. Similarly, Zhou et al. fabricated a polymer electrolyte through the in situ copolymerization of lithium 1-[3-(methacry-loyloxy)propylsulfonyl]-1-(trifluoromethanesulfonyl)imide (LiMTFSI) and pentaerythritol tetraacrylate (PETEA) monomers and then integrating them with poly(3,3-dimethacrylic acid lithium) (PDAAli) coated by glass fiber membranes [85]. This multilayer structured polymer electrolyte possessed an excellent ionic conductivity and great mechanical strength. What is more, it allows almost exclusively Li^+^ transport, which caused the high capacity and long-cycle stability of lithium batteries. 

A novel self-healing polymer electrolyte was reported by Pauline Jaumaux et al. to improve the cyclability and safety of lithium metal batteries [86]. This polymer electrolyte was synthesized by combining a 2-(3-(6-methyl-4-oxo-1,4-dihydropyrimidin-2-yl)ureido)ethyl methacrylate (UPyMA)–pentaerythritol tetraacrylate (PETEA) copolymer matrix with a deep eutectic solvent (DES)-based electrolyte with fluoroethylene carbonate (FEC) as an additive. FEC additives increased the electrochemical stability and ionic conductivity of the electrolyte, and they form a strong fluorine-containing SEI to suppress the irregular growth of dendrites.

Cui et al. utilized polyethylene oxide/lithium bis(trifluoromethanesulfonyl)imide (PEO/LiTFSI) as additives to nanoporous ultrathin polyimide (PI) film to assemble a solid polymer electrolyte [87]. Figure 9g shows the design principle of the polymer–polymer composite solid electrolyte. PI has flame retardancy and ultra-high mechanical strength that inhibits the growth of lithium dendrites. The PEO/LiTFSI in the pores provides ionic conductivity. As shown in Figure 9h, the PI/PEO/LiTFSI all-solid-state lithium-ion batteries exhibited excellent cycling (200 cycles at 0.5 C rate) performance at 60 °C.
Figure 9(**a**) Schematic illustration of the as-prepared HSE structure. (**b**) Capacity and Columbic efficiency versus cycle number of the LTO|PEO/PEG-3LGPS|Li cell and LTO|PEO|Li cell at 0.5 °C. (**c**) Capacity and Columbic efficiency versus cycle number of the LFP|PEO/PEG-3LGPS|Li cell at 0.05 °C, 0.1 °C, and 0.5 °C. Reproduced with permission [82]. Copyright 2020, John Wiley and Sons. (**d**) Stacking model of DLPSE in an all-solid-state cell and molecular structure of PMA. (**e**) The cross-section images of a Li/PEO–LiTFSI/PMA–LiTFSI/LCO cell. (**f**) The cross-section images of a Li/PEO-LiTFSI/LCO cell. Reproduced with permission [84]. Copyright 2019, John Wiley and Sons. (**g**) Schematic showing the design principles of polymer–polymer composite solid-state electrolytes. (**h**) Capacity retention performance of Li/PI/PEO/LiTFSI/LFP and Li/PEO/LiTFSI/LFP cells at 0.5 °C, cycled at 60 °C [87]. Reproduced with permission. Copyright 2019, Springer Nature.
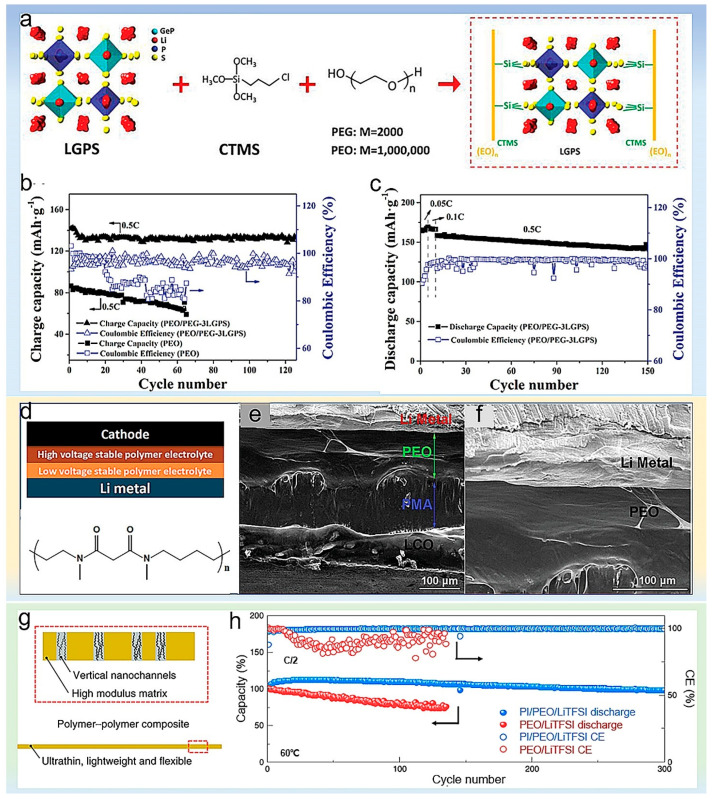



### 4.2. Polymer-in-Salt Electrolyte

For the “salt-in-polymer” electrolytes, the polymer matrices contain relatively fewer lithium salts, resulting in poorer ionic conductivities. By contrast, the Li^+^ ion transportation capability of polymer electrolytes can be significantly enhanced by increasing contents of lithium salts to over 50 wt %, which is normally referred to as “polymer-in-salt” electrolyte [88].

Liu et al. used polyvinylidene fluoride-hexafluoropropylene (PVDF-HFP) and excessive LiTFSI to prepare polymers-in-salt electrolytes (PVHLi-1.1) [89]. The schematic diagram of the Li^+^ ion transport pathway in PISSE is shown in Figure 10a. This electrolyte improves the stability of electrode/electrolyte interface, provides more channels and sites for lithium-ions transportation, and the size effect of the 3D structure can inhibit the irregular growth of lithium dendrites. The SEM image of PVHLi-1.1 film is shown in Figure 10b. PVHLi-1.1 fully infiltrated into the 3D porous TiO_2_/Li, exhibiting high ionic conductivity (1.24 × 10^−4^ S cm^−1^) and high Li^+^ transference number. Finally, the capacity retention rate of polymers-in-salt electrolytes was as high as 86.8% after 800 cycles at 0.2 C (Figure 10c).

Zhao et al. developed a high-concentration lithium-salt polymer electrolyte consisting of poly(ethylene carbonate) (PEC), poly(vinylidene fluoride-co-hexafluoropropylene) (PVDF-HFP) and lithium bis (trifluoromethane sulfonimide) (LiTFSI) [90]. This polymer electrolyte has two transport pathways for lithium ions. One is originated from the PECLi phase, in which lithium salt can be dissolved and dissociated in PEC, forming a “polymer in salt” in PVDF-HFP substrate. The other is that PVDF-HFP and PEC act synergistically to facilitate the Li^+^ movement. As a result, the high-concentration lithium-salt polymer electrolyte presents a high ionic conductivity (1.08 × 10^−4^ S cm^−1^). It shows a wide and stable electrochemical window at around 4.5 V (vs. Li^+^/Li). It is also capable of suppressing the growth of lithium dendrite. 

After that, He et al. fabricated a polymer electrolyte consisting of 80 wt % lithium bis(trifluoromethanesulfonyl)imide (LiTFSI), 20% polyethylene carbonate (PEC), and polyamide (PA) fiber membrane skeleton [91]. The polymer electrolyte exhibits good mechanical strength and high ionic conductivity as well as good wettability to porous electrodes. Wu et al. prepared a polymer-in-salt electrolyte with enhanced oxidative stability by mixing polyethylene oxide (PEO) and lithium bis(fluorosulfonyl)imide (LiFSI) with a dry ball milling [92]. The high voltage stability and ionic conductivity of PEO-based polymer electrolyte systems change with the content of LiFSI. The polymer electrolyte with EO/Li = 1 has high ionic conductivity and high oxidative stability. The first principles show that the coordination of EO with Li^+^ in the electrolyte with EO/Li = 1 reduces the HOMO energy level, thereby increasing the oxidation voltage. As a consequence, the lithium metal batteries with the polymer-in-salt electrolyte delivered a high capacity retention of 74.4% over 186 cycles at a high voltage of 4.3 V.

Chen et al. prepared a salt polymer composed of lithium bis (trifluoromethane sulfonimide) (LiTFSI), polyvinylidene fluoride (PVDF) and double-grafted polysiloxane copolymer by the solution-casting method, and then, they compounded it with cellulose acetate to prepare polymer electrolyte [93]. Figure 10d illustrates the fabrication process of the polymer composite electrolyte membrane by the solution-casting technique strategy schematically. The solid electrolyte exhibits good mechanical properties, which is provided by cellulose acetate, as can been seen in Figure 10e. Polysiloxane, LiTFSI, and the PVDF-prepared polymer salt solid phase extracts offer high ionic conductivity. Consequently, lithium-sulfur batteries assembled with this polymer electrolyte present good cycling performance at 1 °C, as shown in Figure 10f.
Figure 10(**a**) Illustration of Li ion transport pathways in PISSEs. (**b**) SEM image of the PVHLi-1.1 membrane. (**c**) Cycling performance at 0.2 °C. Reproduced with permission [89]. Copyright 2021, John Wiley and Sons. (**d**) Schematic illustration of fabricating composite polymer electrolyte membrane by solution-casting technique. (**e**) Optical photo of 90% (BPSO-150% LiTFSI)-10% PVDF + CA membranes. (**f**) Cycling performance of all-solid-state MCNT@S|90% (BPSO-150% LiTFSI)-10% PVDF + CA|Li cells at 25 °C 1 C rate [93]. Reproduced with permission. Copyright 2018, Elsevier.
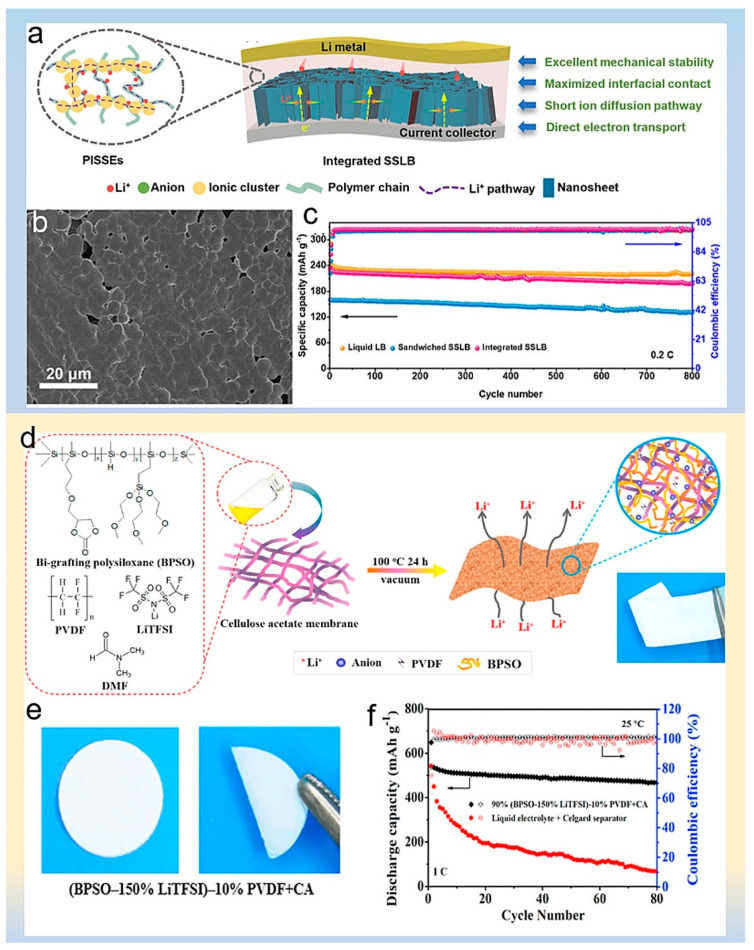



### 4.3. Single-Ion Polymer Electrolyte

In addition to Li^+^ ionic conductivity, Li^+^ ion transference number (*t*_Li_^+^) also plays an indispensable role in affecting the electrochemical properties of polymer electrolytes. When a polymer electrolyte owns a high *t*_Li_^+^, the movement of anions is restrained in the polymer electrolyte, and the derived concentration polarization can also be inhibited. Both the above “polymer-in-salt” and “salt-in-polymer” electrolytes demonstrate low *t*_Li_^+^ values. Therefore, in recent years, it is of high urgency to develop “single-ion-conducting” polymer-type electrolytes with higher *t*_Li_^+^ value [94,95,96].

Porcarelli et al. prepared a single-ion polymer electrolyte by free radical copolymerization [97]. The polymer electrolyte contains lithium 1-[3-(methacryloyloxy)-propylsulfonyl]-1-(trifluoromethylsulfonyl)imide (LiMTFSI), polyethylene glycol methyl ether methacrylate (PEGM), bifunctional polyethylene glycol methyl ether dimethacrylate (PEGDM) and propylene carbonate (PC), as shown in Figure 11a. This single-ion polymer electrolyte exhibits a high Li^+^ transference number and excellent ionic conductivity (σ ≈ 10^−4^ S cm^−1^). It also exhibits good mechanical strength, low relaxation temperature (T_g_), good thermal stability and a wide voltage window. Similarly, Nguyen et al. developed self-supporting nanostructured multiblock copolymerized poly(arylene ether sulfone) (PES) single-ionic polymer electrolytes [98], which ensures the synergy of key performance, mechanical stability and ion mobility to be maintained over the entire temperature range, even in the existence of high dielectric and thermally stable ECs. Borzutzki et al. introduced homopolymers containing polysulfone amide moieties on the polymer backbone to form single-ion conducting polymers containing a large number of –C(CF_3_)_2_ groups [99]. 

A 3D porous structural polymer electrolyte was constructed by Li_0.33_La_0.557_TiO_3_ nanowires (LLTO) and bacterial cellulose (BC) by Ding et al., which exhibited excellent affinity with liquid electrolytes [100]. Figure 11b illustrates the preparing process of the multifunctional BC/LLTO CGE matrix schematically. The polymer electrolyte exhibits excellent mechanical strength and a high lithium-ion migration number. It also exhibits good thermal stability, as shown in Figure 11c. The synergistic effect of LLTO and robust BC framework enables the effective suppression of Li dendrites growth and stable Li^+^ deposition. The Li||BC/LLTO||LiFePO4 cell delivered ultra-high capacity retention (98.5%) and reversible discharge capacity (151 mAh g^−1^) after 100 cycles.

In addition to the above-mentioned polymer-in-salt, salt-in-polymer, and single-ion polymer electrolytes, functional materials such as metal–organic frameworks (MOFs) are being increasingly explored as additives in polymer electrolytes. The high specific surface area of MOFs can generate steric effects and vacant coordination sites, which can facilitate Li^+^ transport and enhance lithium storage kinetics in LMBs [101,102].

In summary, the utilization of polymer electrolytes instead of conventional liquid electrolytes can facilitate uniform Li deposition and suppress Li dendrite growth and thus improve the safety and prolong the lifespan of LMBs. An ideal polymer electrolyte for lithium metal batteries should have good mechanical strength, high ionic conductivity, certain flexibility to ensure good contact at the electrode/electrolyte interface, and abundant surface functionalities for the efficient regulation of Li^+^ flux. Bio-derived polymers as well as re-processable and degradable polymers are environmentally friendly and sustainable, and they are the future development trends of polymer electrolytes. 

## 5. Conclusions and Perspectives

Polymer materials are playing crucial roles in LMBs, varying from polymeric artificial SEI and polymeric functional interlayer to polymer electrolyte. Thus, engineering polymer architectures and functionalities is of great significance for constructing high-energy and long-life LMBs.

Polymeric artificial SEIs have been constructed to enhance the surface/interfacial stability of lithium metal anodes. They should possess favorable mechanical and electrochemical properties toward high-performance LMBs. High viscoelasticity and large shear strength are desirable for accommodating the large volume fluctuation over a prolonged cycling process. Polymeric artificial SEI should possess both high Li^+^ ionic conductivity and large *t*_Li_^+^ to facilitate rapid and smooth Li^+^ transportation. Inorganic additives can also be introduced to further enhance the mechanical and electrochemical properties of polymer matrices.

Polymeric interlayers are functional films between the separator and lithium metal anode to induce uniform Li deposition and inhibit Li dendrite growth. They can exist as free-standing layers on top of lithium metal anode or be attached on the bottom of the separator. A single polymer can work as a functional layer alone, or multiple polymers can work together, utilizing the synergy of different components. In both cases, careful choices of suitable surface functionalities are necessary to tune the lithium ion flux at the electrode surface. In inorganic/polymer composite interlayers, the inorganic components can bring about numerous favorable characteristics, including high mechanical strength, good ionic conductivity, excellent shape compliance, and good compatibility with lithium metal anodes.

Polymer electrolytes have attracted extensive research attention in the past few decades. They demonstrate favorable flexibility, desirable Li^+^ ion conductivity, and good compatibility with both cathode and lithium metal anode. Boosted cycling stability can thus be achieved in LMBs after careful choice of proper polymer electrolytes. Most polymer electrolytes are “salt-in-polymer” electrolytes with lithium salt contents lower than 50 wt %. The electrochemical properties can be well enhanced by functional units adjustment toward high ionic conductivities and wide electrochemical stability windows. In “polymer-in-salt” electrolytes, lithium salt contents are higher than 50 wt % with correspondingly improved *t*_Li_^+^. The potential polarization of the battery and side reactions between electrolyte and electrode can be reduced. In “single-ion-conducting” polymer electrolytes, a high *t*_Li_^+^ is favorable for rapid cation transportation and high-power capabilities of LMBs.

More efforts that could promote the development of novel polymer materials in LMBs toward practical applications can be made in the following directions:(1)For polymeric artificial SEI and interlayers, multifunctional polymers should be synthesized to concurrently achieve rapid Li^+^ transportation, high mechanical modulus, and good compatibility with lithium metal anode. The functional layers should possess optimized thickness: for one thing to minimize the mass of inactive components in LMB and for another to assure moderate mechanical strength.(2)For polymer electrolyte, a high Li^+^ conductivity is a prerequisite for its proper function. A prospective approach is decoupling Li^+^ conduction from the polymer’s segmental motion. The thickness of polymer electrolyte should also be well adjusted to balance the Li^+^ ionic conductivity and mechanical property. Moreover, it is urgent to enhance the compatibility of polymer electrolytes with highly reductive lithium metal anode and oxidative high-voltage cathode.(3)Exploring facile and low-cost synthetic procedures for functional polymers is of great significance for mass production.(4)Advanced characterization tools must be used to observe the evolution of ASEIs during cycling to guide the design of better coatings. Advanced characterization techniques should be utilized to in situ monitor lithium metal deposition at electrode interfaces, thus unveiling the mechanism of Li metal nucleation/growth control by functional polymers.(5)Battery testing at practical conditions is also required to facilitate practical applications of functional polymers in LMBs. Cell fabrication parameters such as the amount of electrolyte used, cathode mass loading and external pressure applied could affect the electrochemical performances of LMBs. Although a massive electrolyte could sustain a longer lifespan, it sacrifices the energy density of the battery cell; high cathode mass loading is necessary for achieving high energy density of batteries, yet the derived high internal resistance should be of careful consideration; a certain external pressure is favorable for inducing more uniform Li deposition and thus boosting the cycle life [103], while too high external pressure is infeasible in practical scenarios. Therefore, battery testing should be performed with moderate electrolyte amount, cathode mass loading as well as external pressure to more rationally evaluate the functional polymer-enhanced electrochemical performances in LMBs.

## Figures and Tables

**Figure 1 polymers-14-03452-f001:**
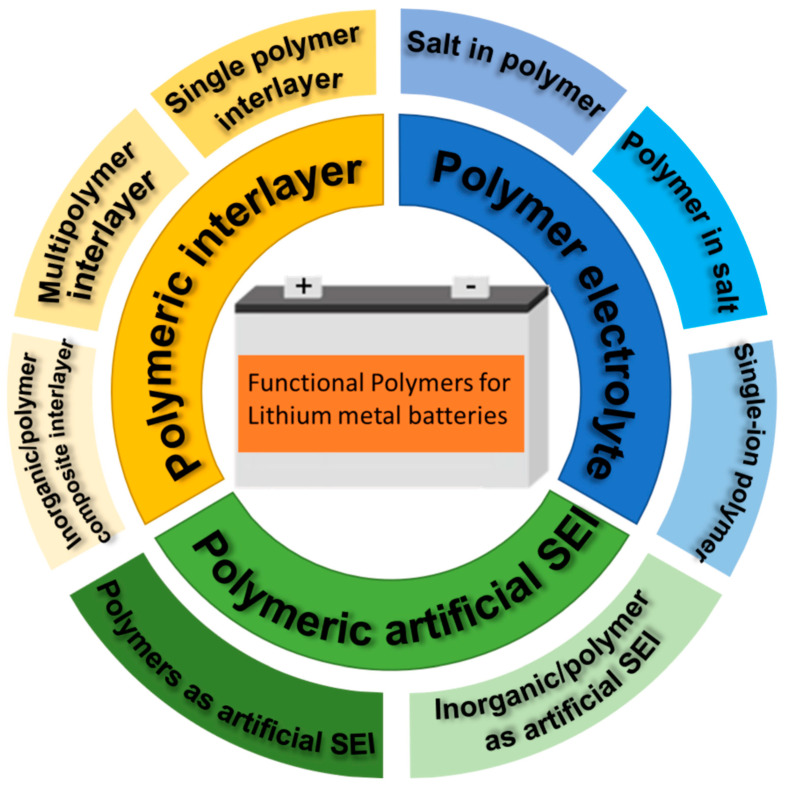
Classification of functional polymer materials in LMBs.

**Figure 4 polymers-14-03452-f004:**
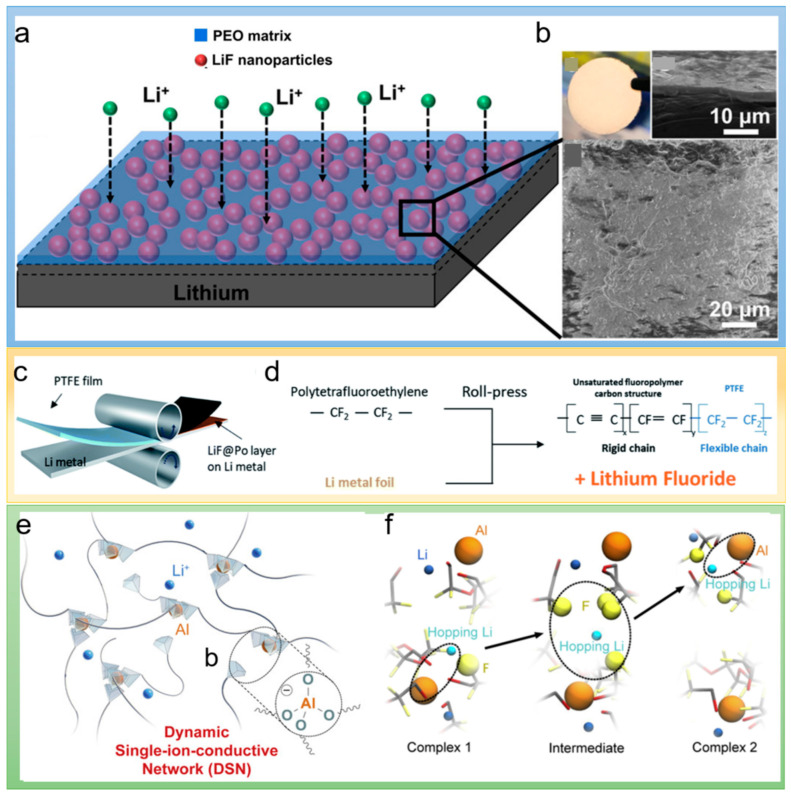
(**a**) Scheme of the LiF/PEO composite coating on lithium metal. (**b**) Photograph of the lithium metal electrode coated with the LiF/PEO composite; top view and cross-sectional SEM images of the LiF/PEO composite coating [48]. Reproduced with permission. Copyright 2020, American Chemical Society. (**c**) Schematic illustration of the LiF or LiF@Po protection layer formation on the Li metal by the roll pressing process between Li metal and the PTFE film. (**d**) Chemical reaction model during the roll pressing process. Reproduced with permission [49]. Copyright 2020, Royal Chemical Society. (**e**) Schematic diagram of the molecular structure of the dynamic single-ion conducting network (DSN). (**f**) Li^+^ ion transport pathway. The hopping Li^+^ ion is shown in light blue while irrelevant Li^+^ ions are dark blue. F atoms within 3 A° of the hopping Li are emphasized with yellow spheres. For clarity, FTEG chains are faded [50]. Reproduced with permission. Copyright 2019, Elsevier.

**Figure 6 polymers-14-03452-f006:**
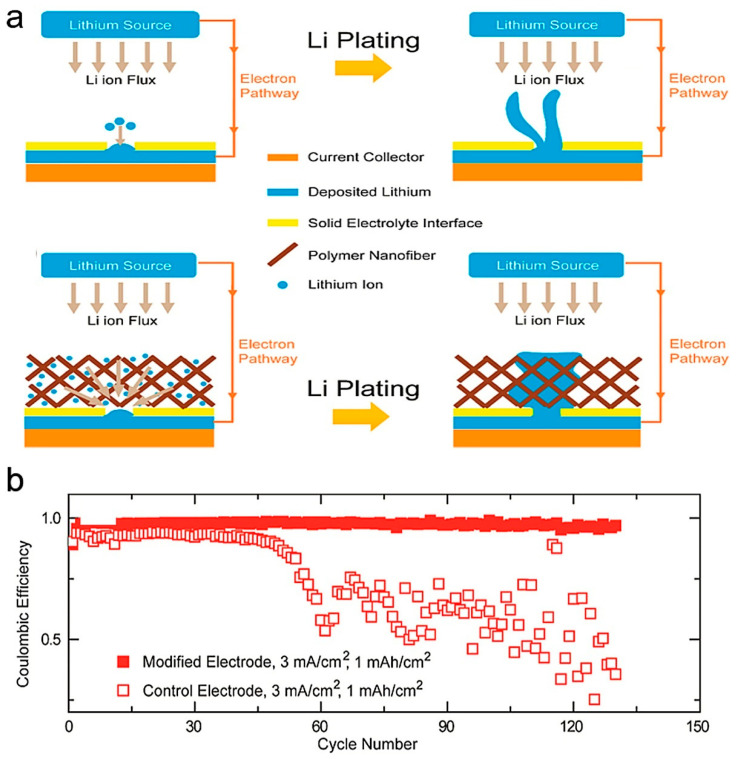
(**a**) Schematic illustration of lithium deposition on bare copper electrode and polymer nanofiber network modified Cu electrode. (**b**) Comparison of the Coulombic efficiency of Li deposition on bare Cu and Cu-OxPAN electrode with current densities of 3 mA cm^−2^. The amount of Li plated in each cycle is 1 mAh cm^−2^. Reproduced with permission [62]. Copyright 2015, American Chemical Society.

**Figure 7 polymers-14-03452-f007:**
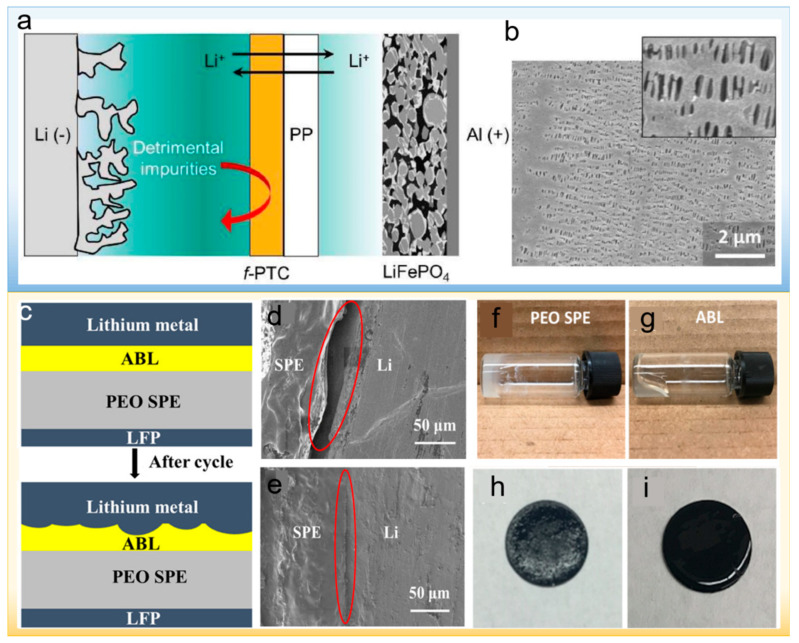
(**a**) A schematic for the suppressed crossover effect observed in the typical Li metal-based full cells by the addition of *f*-PTC. (**b**) SEM images (inset: magnified surface image) of and cycled PP laminated with *f*-PTC harvested from the Li||LiFePO_4_ full cell [69]. Reproduced with permission. Copyright 2022, Elsevier (**c**) Schematic of the interface contact between Li metal anode and SPE before and after repeated cycles with ABL. (**d**,**e**) Cross-sectional SEM images of symmetric cells (Li/SPE/Li) after being charged to 20 mAh cm^2^ at 0.04 mA cm^2^ current density. (**d**) Without and (**e**) with ABL. The working temperature was set as 50 °C for all batteries. Reproduced with permission. The completely dried (**f**) PEO SPE. (**g**) ABL. The surface condition of the LFP cathode with PEO SPE (**h**) without and (**i**) with ABL [72]. Copyright 2019, American Chemical Society.

**Figure 8 polymers-14-03452-f008:**
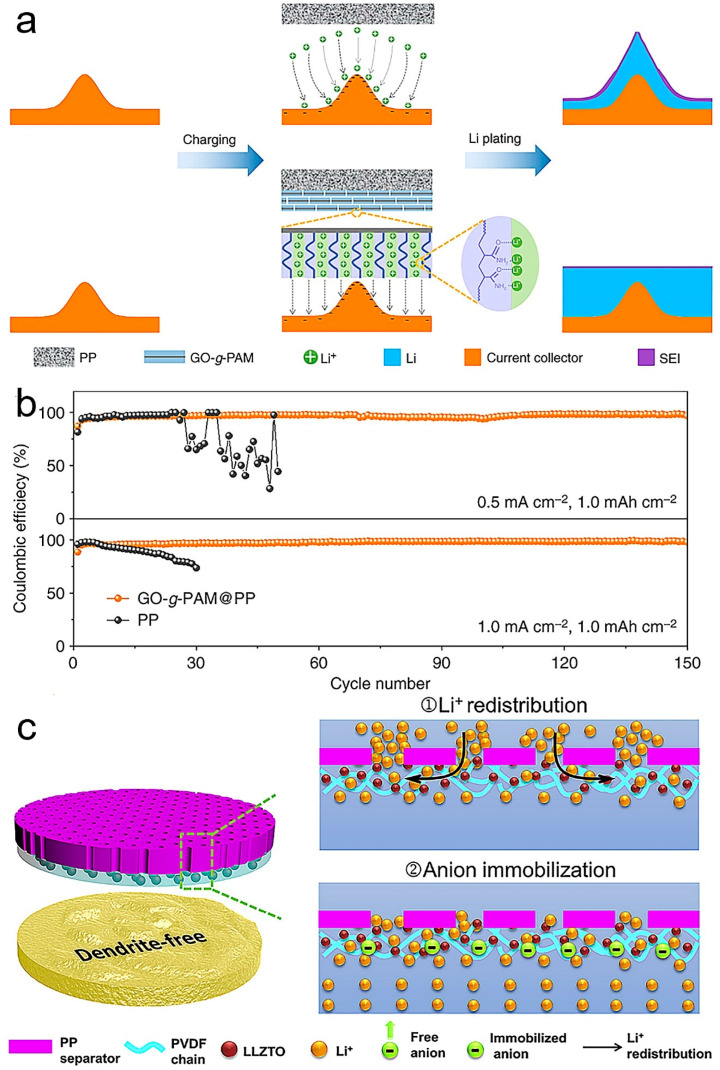
(**a**) Schematic diagram of Li deposition on electrodes with microscopic surface roughness. Schematics showing Li deposition on an electrode with a PP separator (top). Schematics showing Li deposition on an electrode with a GO-*g*-PAM@PP separator (bottom). (**b**) Coulombic efficiencies of Li-Cu cells with GO-*g*-PAM@PP and PP separators with a cycling capacity of 1 mAh  cm^−2^ at various current densities [75]. Reproduced with permission. Copyright 2019, Springer Nature. (**c**) Schematic illustration of the Li^+^ deposition behaviors through anion-immobilized PP@PLLZ separator with multiple highly-conductive ion pathway. Reproduced with permission [77]. Copyright 2020, Elsevier.

**Figure 11 polymers-14-03452-f011:**
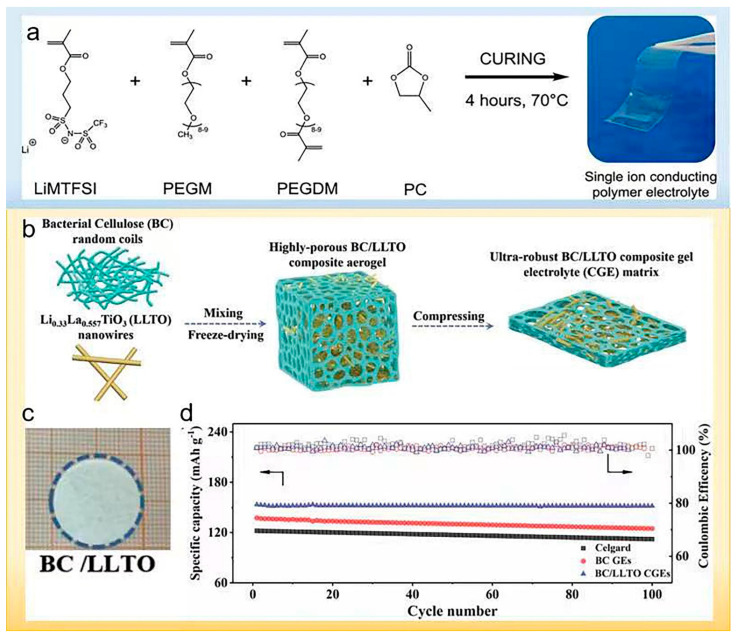
(**a**) General pathway for the preparation of SIPE films. Reproduced with permission [97]. Copyright 2016, American Chemical Society. (**b**) Schematic illustration of the fabrication of multifunctional BC/LLTO CGE matrix. (**c**) Optical photograph of thermal shrinkage BC/LLTO CGE matrix after exposure to 150 °C for 2 h. (**d**) Cycle performance of BC/LLTO CGEs at 0.2 C as compared with the counterparts [100]. Reproduced with permission. Copyright 2020, John Wiley and Sons.

**Table 1 polymers-14-03452-t001:** Summary of polymer interlayer thicknesses in reported references.

Polymeric Interlayers	Thickness	Ref.
CPC	2.5 µm	[64]
CTS-PEO-PTEGDMA	25 nm	[68]
ABL	10 µm	[69]
NSP	2 µm	[70]
PCI	5 µm	[72]
PLLZ	10 µm	[74]

## Data Availability

The data presented in this study are available on request from the corresponding author.

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
