# Peer review of "Functional Polymer Materials for Advanced Lithium Metal Batteries: A Review and Perspective"

_polymers, 2022, doi:10.3390/polym14173452_

Round 1
Reviewer 1 Report
Summary:
The Review Article polymers 1868487 titled, “Functional Polymer Materials for Advanced Lithium Metal Batteries: A Review and Perspective,” makes a good summary of the material, the design, the development of lithium metal batteries with the modification of various functional polymers as the artificial solid electrolyte interface, the functional interlayers, and the polymer electrolyte between a separator and the lithium metal anode to improve the electrochemical stability of lithium metal anodes and the resulting lithium metal batteries. With the case study, the relationship between functionalized polymer materials and improved electrochemical performance will be elucidated, which showcases the current important research development and the future challenges. To address the future issues, four key perspectives are suggested.
General comment:
This review articles summarizes the recent important research progresses in using polymers for the advanced lithium metal batteries. With a support of 94 references, the manuscript is organized and highlights the importance of polymer materials in the lithium metal protection, stabilization, and utilization. A minor revision is therefore suggested to provide some necessary data and parameters. Hope the authors feel the comment useful.
Comments:
(1) The review article gives a comprehensive paper survey of recent progresses in the polymer materials for the lithium metal anodes. The performance improvements and key factors are reported. It is suggested to summarize or comment on the cell preparation and cell fabrication parameters.
[Suggestion] Please consider commenting or summarizing the cell preparation and cell fabrication parameters.
(2) The use and the application of the polymer materials are highlighted and categorized for the future researchers. Accordingly, it is suggested to arrange the polymers used, if it is possible.
[Suggestion] Please consider arranging the polymers reported.
(3) The dimension data of the polymer interlayers and the polymer electrolytes are suggested to be reviewed and summarized for a better understanding.
[Suggestion] Please consider commenting or summarizing the dimensions of the polymer interlayers and polymer electrolytes
(4) Some recent reference using polymer materials for high-energy-density lithium metal batteries are suggested (J. Electrochem. Soc. 2021, 168, 090501, doi: 10.1149/1945-7111/ac1cc3; Polymers 2021, 13, 535, doi: 10.3390/polym13040535; J. Mater. Chem. A 2022, 10, 13719-13726, doi: 10.1039/D2TA01867E).
[Suggestion] Please support the discussion with reference citation and summarize the development trend.
(5) Please review and proofread the manuscript. Please check the resolution of the images and plots.
Author Response
The Review Article polymers 1868487 titled, “Functional Polymer Materials for Advanced Lithium Metal Batteries: A Review and Perspective,” makes a good summary of the material, the design, the development of lithium metal batteries with the modification of various functional polymers as the artificial solid electrolyte interface, the functional interlayers, and the polymer electrolyte between a separator and the lithium metal anode to improve the electrochemical stability of lithium metal anodes and the resulting lithium metal batteries. With the case study, the relationship between functionalized polymer materials and improved electrochemical performance will be elucidated, which showcases the current important research development and the future challenges. To address the future issues, four key perspectives are suggested.
General comment:
This review articles summarizes the recent important research progresses in using polymers for the advanced lithium metal batteries. With a support of 94 references, the manuscript is organized and highlights the importance of polymer materials in the lithium metal protection, stabilization, and utilization. A minor revision is therefore suggested to provide some necessary data and parameters. Hope the authors feel the comment useful.
Reply: We thank the reviewer for the positive evaluation of our manuscript and the valuable comments.
Comments:
(1) The review article gives a comprehensive paper survey of recent progresses in the polymer materials for the lithium metal anodes. The performance improvements and key factors are reported. It is suggested to summarize or comment on the cell preparation and cell fabrication parameters.
[Suggestion] Please consider commenting or summarizing the cell preparation and cell fabrication parameters.
Reply: We thank the reviewer for the valuable suggestion. The effect of cell fabrication parameters on LMB performances have been summarized and added in the revised Conclusion section (highlighted in red).
(2) The use and the application of the polymer materials are highlighted and categorized for the future researchers. Accordingly, it is suggested to arrange the polymers used, if it is possible.
[Suggestion] Please consider arranging the polymers reported.
Reply: Thanks for the reviewer’s comment. We have arranged the polymers reported in the manuscript prior to the Introduction section.
(3) The dimension data of the polymer interlayers and the polymer electrolytes are suggested to be reviewed and summarized for a better understanding.
[Suggestion] Please consider commenting or summarizing the dimensions of the polymer interlayers and polymer electrolytes
Reply: Thanks for the reviewer’s important suggestion. We have stated the specific thickness range of the interlayer in Section 3 and summarized interlayer thickness in Table 1.
(4) Some recent reference using polymer materials for high-energy-density lithium metal batteries are suggested (J. Electrochem. Soc. 2021, 168, 090501, doi: 10.1149/1945-7111/ac1cc3; Polymers 2021, 13, 535, doi: 10.3390/polym13040535; J. Mater. Chem. A 2022, 10, 13719-13726, doi: 10.1039/D2TA01867E).
[Suggestion] Please support the discussion with reference citation and summarize the development trend.
Reply: We have cited the three important papers in the revised manuscript. Besides, the development trend has been also summarized and added (highlighted in red).
(5) Please review and proofread the manuscript. Please check the resolution of the images and plots.
Reply: We have carefully reviewed the manuscript and checked the resolution of the images and plots.

Reviewer 2 Report
I have read the review article entitled “Functional Polymer Materials for Advanced Lithium Metal Batteries” by X. Yu et al. Rechargeable batteries that use Li metal as an anode are regarded as the most viable alternative to state-of-the-art lithium-ion batteries (LIBs). This is mainly because metallic lithium used in Li metal batteries (LMBs) is capable of higher-density energy storage. The concept of using metallic Li itself is not new as LMB is the earliest prototype of lithium secondary batteries, including the device originally developed by Whittingham. An LMB uses a thin layer of lithium foil as the anode, aprotic organic lithium solution as the electrolyte, and lithiated layered metal oxides in the presence of binders, as the cathode. The formation of gradients in the concentration of lithium ions promoting the growth of dendrites is the major obstacle in LMB. To address and overcome these issues, the authors have presented their review and perspective on the novel polymer material to enhance the performance of LMBs. Therefore, the topic of review with an interlink of electrochemical performances to metallic Li-ion batteries is of interest and suitable for the chosen journal Polymers, MDPI.
The authors should not distract the readers by going away from the said objectives of the presented review. Occasionally, this has occurred in this review. I have commented below. Sections 2- 3 should be provided with a nice introduction and a stand-alone summary for that respective section.
Some parts of the submitted review work must be improved.
Following are my general and specific comments:
· What is the difference “Salt in Polymer” versus “Polymer in Salt” shown in Figure 1?
· Page 2; prior to section 2. Please provide the general characteristics of polymer/solid/Gel polymer electrolytes and polymer-based artificial SEI.
· Page 3; “Biopolymers are natural polymers” in addition to agarose, alginates (doi.org/10.1021/acsaem.1c01111), and chitosan (10.1039/C5DT00622H; 10.1039/C6NJ00521G) are also widely used for these purposes. Please include and discuss.
· Section 2.2 – are inorganic additives equivalent to non-conductive/reinforcing fillers? In this context, alkoxy silane terminated oligomeric PEO-OH were also reported to graft PEO oligomers onto SiO2 nanoparticles in water.
· Metal-organic frameworks are increasingly explored as additives for polymer electrolytes, any comments?
· The replacement of conventional liquid electrolytes in LMBs with polymer electrolytes can improve safety. Arguably, how about the cycling life by contributing to regulating the morphology of Li deposits?
· Bio-derived polymers, such as cellulose, as well as reprocessable and degradable polymers, toward more sustainable electrolytes is the future for LMBs. So, stress this under a separate section.
· Maybe a strategy to achieve stable LMBs that employ traditional liquid electrolytes consists of engineering the structure of polymer separators, any comment/discussion on this area?
· An effective conclusion summarising the ideal electrolytes for LMBs along with the strategy to prevent dendrite growth and eliminate the hazards associated with the commercial volatile and flammable liquid electrolytes need to be given. Passivation layer on cycling needs to be addressed through any additives.
· The clarity of the figures can be improved.
· Section 5 – conclusion has been provided but there is no perspectives from author’s view.
Author Response
I have read the review article entitled “Functional Polymer Materials for Advanced Lithium Metal Batteries” by X. Yu et al. Rechargeable batteries that use Li metal as an anode are regarded as the most viable alternative to state-of-the-art lithium-ion batteries (LIBs). This is mainly because metallic lithium used in Li metal batteries (LMBs) is capable of higher-density energy storage. The concept of using metallic Li itself is not new as LMB is the earliest prototype of lithium secondary batteries, including the device originally developed by Whittingham. An LMB uses a thin layer of lithium foil as the anode, aprotic organic lithium solution as the electrolyte, and lithiated layered metal oxides in the presence of binders, as the cathode. The formation of gradients in the concentration of lithium ions promoting the growth of dendrites is the major obstacle in LMB. To address and overcome these issues, the authors have presented their review and perspective on the novel polymer material to enhance the performance of LMBs. Therefore, the topic of review with an interlink of electrochemical performances to metallic Li-ion batteries is of interest and suitable for the chosen journal Polymers, MDPI.
The authors should not distract the readers by going away from the said objectives of the presented review. Occasionally, this has occurred in this review. I have commented below. Sections 2- 3 should be provided with a nice introduction and a stand-alone summary for that respective section.
Reply: We thank the reviewer for the positive evaluation of our manuscript and the valuable comments.
Some parts of the submitted review work must be improved.
Following are my general and specific comments:
What is the difference “Salt in Polymer” versus “Polymer in Salt” shown in Figure 1?
Reply: The definitions of the two types of polymer electrolytes has been defined in 4.2. "Polymer in salt" and "salt in polymer" are distinguished according to the mass content of lithium salt in the polymer electrolyte. When lithium salt content exceeds 50%, it is defined as "polymer in salt". And a "salt in polymer" electrolyte refers to a polymer matrix containing less than 50% lithium salt.
Page 2; prior to section 2. Please provide the general characteristics of polymer/solid/Gel polymer electrolytes and polymer-based artificial SEI.
Reply: Thanks for the reviewer’s suggestion. We have supplemented the general characteristics of the polymer artificial SEI, polymer interlayer and polymer electrolyte at the end of the Introduction section in the revised manuscript (highlighted in red).
- Page 3; “Biopolymers are natural polymers” in addition to agarose, alginates(doi.org/10.1021/acsaem.1c01111), and chitosan (10.1039/C5DT00622H; 10.1039/C6NJ00521G) are also widely used for these purposes. Please include and discuss.
Reply: We have added the suggested references and discussed their applications in the section of biopolymers in the revised manuscript.
- Section 2.2 – are inorganic additives equivalent to non-conductive/reinforcing fillers? In this context, alkoxy silane terminated oligomeric PEO-OH were also reported to graft PEO oligomers onto SiO2 nanoparticles in water.
Reply: Inorganic additives are not fully equivalent to non-conductive/reinforcing fillers. For some inorganic additives with high ionic conductivities, they not only enhance the mechanical strength, but also improve the ion conducting capability of the composite.
- Metal-organic frameworks are increasingly explored as additives for polymer electrolytes, any comments?
Reply: We thank the reviewer for the valuable comment. And we have summarized the typical features of MOFs and their unique functions in polymer electrolytes at the end of Section 4 in the revised manuscript (highlighted in red).
The replacement of conventional liquid electrolytes in LMBs with polymer electrolytes can improve safety. Arguably, how about the cycling life by contributing to regulating the morphology of Li deposits?
Reply: Polymer electrolyte can improve the safety of LMBs because of their high mechanical strength to inhibit Li dendrites and also high flexibility to accommodate the large volume fluctuation upon charge/discharge. Moreover, some polymer electrolytes have abundant surface functionalities to enable regulation of Li+ flux at the electrode/electrolyte interface, which has been discussed in the revised manuscript.
- Bio-derived polymers, such as cellulose, as well as reprocessable and degradable polymers, toward more sustainable electrolytes is the future for LMBs. So, stress this under a separate section.
Reply: We thank the reviewer for this important suggestion. We have highlighted this point at the end of Section 4 in the revised manuscript (highlighted in red).
Maybe a strategy to achieve stable LMBs that employ traditional liquid electrolytes consists of engineering the structure of polymer separators, any comment/discussion on this area?
Reply: Thank you for the reviewer's valuable comment. In this paper, we mainly highlight the functional polymer design to regulate Li deposition at the electrode/electrolyte interface. We agree with reviewer that engineering the structure of separators are very important for improving the cycle life of LMBs. One typical example is the modification by functional films on separators (categorized as functional interlayers in this paper). Besides, mechanical strength enhancement is also very important to suppress the Li dendrite growth (DOI: 10.1016/j.cej.2020.124258; DOI: 10.1039/c8ta11795k). However, this is not the focus of this review paper, therefore it is not highlighted in the manuscript.
An effective conclusion summarising the ideal electrolytes for LMBs along with the strategy to prevent dendrite growth and eliminate the hazards associated with the commercial volatile and flammable liquid electrolytes need to be given. Passivation layer on cycling needs to be addressed through any additives.
Reply: We agree with the reviewer that optimized electrolyte formulations are quite important for preventing dendrite growth and eliminating the hazards associated with the commercial volatile and flammable liquid electrolytes. Ideal electrolyte formulations including functional additives to remove passivation layers have been discussed in another recent review paper published by us (Energy Environ. Mater., 2022, doi.org/10.1002/eem2.12355), which we think are not the focuses of this paper. Therefore, they are not discussed here.
The clarity of the figures can be improved.
Reply: The resolutions of the images have been improved.
Section 5 – conclusion has been provided but there is no perspectives from author’s view.
Reply: We have shown perspectives in Section 5.

Round 2
Reviewer 2 Report
In this reviewer’s opinion, the revised version is OK. In which, the authors have taken my queries into a consideration and revised the manuscript. I have read the highlighted parts of the work, it is suitable to publish.